# Nanocrystal Suspensions for Enhancing the Oral Absorption of Albendazole

**DOI:** 10.3390/nano12173032

**Published:** 2022-09-01

**Authors:** Zhiwei Liang, Min Chen, Yuanyuan Yan, Dongmei Chen, Shuyu Xie

**Affiliations:** 1National Reference Laboratory of Veterinary Drug Residues (HZAU), Wuhan 430070, China; 2MAO Key Laboratory for Detection of Veterinary Drug Residues, Huazhong Agricultural University, Wuhan 430070, China; 3MOA Laboratory for Risk Assessment of Quality and Safety of Livestock and Poultry Products, Huazhong Agricultural University, Wuhan 430070, China

**Keywords:** albendazole, nanocrystals, insolubility, bioavailability, oral absorption

## Abstract

Albendazole (ABZ), an effective benzimidazole antiparasitic drug is limited by its poor solubility and oral bioavailability. In order to overcome its disadvantages, ABZ nanocrystals were prepared using a novel bottom-up method based on acid-base neutralization recrystallization with high-speed mixing and dispersing. The cosolvent, stabilizer and preparation temperature were optimized using single factor tests. The physicochemical properties, solubility and pharmacokinetics of the optimal ABZ nanocrystals were evaluated. The high-performance liquid chromatography (HPLC), differential scanning calorimetry (DSC) and X-ray powder diffraction (XRD) showed that ABZ had no structural and crystal phase change after nanocrystallization. The saturated solubility of ABZ nanocrystals in different solvents was increased by 2.2–118 fold. The oral bioavailability of the total active ingredients (ABZ and its metabolites of albendazole sulfoxide (ABZSO) and albendazole sulfone (ABZSO2)) of the nanocrystals in rats was enhanced by 1.40 times compared to the native ABZ. These results suggest that nanocrystals might be a promising way to enhance the solubility and oral bioavailability of ABZ and other insoluble drugs.

## 1. Introduction

Albendazole (ABZ), a broad-spectrum, highly effective and low toxic benzimidazole antiparasitic drug, is widely used for treating hydatid cysts and cerebral cysticercosis [1,2]. It is the first choice for microsporidiosis in acquired immunodeficiency syndrome (AIDS), ascariasis, enterobiasis, hookworm infections and neurocysticercosis [3,4]. It has also been widely used to treat various intestinal and systemic parasitosis in domestic animals and companion animals [5,6]. After entering into the body, it can be oxidized to albendazole sulfoxide (ABZSO) and then further oxidized to albendazole sulfone (ABZSO2) (Figure 1). These two main metabolites have insect repellent activity [7]. However, its efficacy is often limited by poor oral absorption mainly due to its low aqueous solubility, which severely limits its clinical usage. The oral bioavailability of ABZ in dogs is only 12.57% and less than 5% in humans [8]. Therefore, it is very important to improve the solubility of albendazole for its clinical application.

At present, many preparation technologies including liposomes [9], β-cyclodextrin complex [10] and amorphous solid dispersion [11] have been reported to improve the absorption of ABZ, while none of these ABZ preparations have entered the clinical research stage. Nanoparticles with below 1000 nm in size could significantly increase the saturated solubility and dissolution of insoluble drugs by enhancing their surface area and thus improve the oral absorption [12,13,14,15]. Among multiple methods used to improve the solubility of insoluble drugs, nanocrystals has become the preferred strategy [12]. For example, the nanocrystals prepared by our group significantly improved the oral bioavailability of cyadox and oxfendazole [16,17]. Some innovative oral formulations based on nanocrystals have been come to market, and this technology offers more potential in the future [18].

Some authors have used nanocrystals to increase the solubility, dissolution, oral bioavailability and efficacy of albendazole and ricobendazole in infected mice, rats and dogs [19,20,21,22,23,24]. For example, in a recent report, the nanocrystals of ABZ enhanced its solubility by 8.9 fold [22]. A novel nanocrystalline formulation of ABZ obtained by spray drying with Poloxamer 188 as a stabilizer demonstrated a cyst inhibition effect of 3.7-fold greater than that of the commercial ABZ oral product (Albenda) [23]. The nanocrystal-based formulation also improved the pharmacokinetic performance and therapeutic effects of ABZ in dogs [21]. It should be noted that nanocrystals cannot only be delivered by oral routs but also by intravenous injection and other ways according to the clinical requirement [15]. For example, the nanocrystalline formulation of ABZ was intradermally delivered using dissolving microneedles [25]. Lastly, ABZ nanocrystals were included within 3D printed tablets or printlets using a melting solidification 3D printing technique, which allowed a high drug nanocrystal content of up to 50% *w*/*w* [26]. In previous reports, the preparation methods of ABZ nanocrystals included the antisolvent precipitation process [23], high pressure homogenization (HPH), and nanomilling combing with spray drying [26], etc. These methods face high production costs (1:1 for drugs and surfactants), organic solvents (e.g., dichloromethane, dimethyl sulfoxide) and the strict equipment requirements of HPH [21,27], which might be not easily suitable in industrial production for human and veterinary clinics. Thus, there arises a need to find a simple, efficient and scalable method to produce ABZ formulations.

In this study, ABZ nanocrystal suspension was prepared using a novel bottom-up approach based on acid-base neutralization followed by high-speed mixing and dispersing to easily achieve industrial production. The properties, solubility, dissolution and pharmacokinetics of ABZ nanocrystal suspension in rats were evaluated.

## 2. Materials and Methods

### 2.1. Preparation of ABZ Nanocrystal Suspension

The ABZ nanocrystals suspension was prepared using acid-base neutralization combined with high-speed mixing and dispersing. The schematic representation of the ABZ nanocrystals preparation is shown in Figure 2. Three gram ABZ were dissolved in 18 mL solution containing 15 g malic acid at 80 °C under thermostatic magnetic stirrer (Shanghai Respiratory Analytical Instrument Co., Ltd. 90-1, Shanghai, China). When the drug was completely dissolved, the heating button of the thermostatic magnetic stirrer was turned off. Subsequently, 15 mL 1.0 mol/L NaOH solution were slowly added into the acidic aqueous solution containing ABZ under agitation, followed by adding 5% polyvinyl alcohol (PVA) 5 mL. The recrystallized ABZ was homodispersed using a high-speed dispersing device (Laboratory digital display, high speed shear emulsion mixer, JRJ300-SH, Shanghai Hushi Industrial Co., Ltd., Shanghai, China) at the rotational speed of 2000 rpm for 5 min. Finally, the volume was fixed to 50 mL with sterilized water.

### 2.2. Characterization of the ABZ Nanocrystals

#### 2.2.1. Scanning Electron Microscopy (SEM)

The morphology of ABZ nanocrystals was determined using SEM to examine the morphology. The dried nanocrystals sample was placed on a silicon wafer and subsequently sprayed with gold via the sputter coater for 8 min. The sprayed thickness was about 50 nm, and it was then observed using a scanning electron microscope (High-Tech Co., Tokyo, Japan).

#### 2.2.2. Determination of Size, Polydispersity Index (PDI) and Zeta Potential of Nanocrystals

The size, PDI and zeta potential of the nanocrystals were measured using a laser particle size analyzer. The appropriate amount of ABZ nanosuspension was appropriately diluted with double distilled water and measured using a laser particle size analyzer (ZX3600, Malvern, UK) at 20 °C. The diameter was calculated using volume distribution. Three repetitions were determined by using three independent batches.

#### 2.2.3. Differential Scanning Calorimetry (DSC)

The dried ABZ nanocrystals were determined using DSC. The sample was scanned to 300 °C at a scanning rate of 10 °C/min using the 200 pc instrument (NETZSCH, Selbu, Germany). The temperature range included the melting point of the drug and excipients. Inert gas was introduced into the measurement to purify nitrogen at a pressure of 20 psi and a flow rate of 50 mL/min. Each experiment was performed in triplicate.

#### 2.2.4. X-ray Diffraction (XRD) Study

X-ray diffraction analysis was carried out for the bulk drug and for the prepared nanocrystals using X-ray diffractometer (D2 phaser Make: Bruker, Karlsruhe, Germany). The powder was scanned from 5 to 40° (2θ).

#### 2.2.5. Sedimentation Rate, Redispersibility and pH

To determine the sedimentation rate, the original height (H0) of 50 mL nanosuspension in a 100 mL measuring cylinder was recorded after shaking. After standing for 3 h, the height (H) was measured again. The sedimentation rate was calculated according to the following formula:F = H/H0.

The redispersibility of the nanosuspension was measured using redispersion time after its sedimentation. Briefly, 50 mL nanosuspension was placed in a 100 mL measuring cylinder for several days to produce sedimentation, then the layered nanosuspension was shaken under a magnetic oscillator rotating at 20 r/min to calculate its redispersion time.

After sample was prepared, the pH of ABZ nanosuspension was tested with the aid of a pH meter (FE28, Shanghai Mettler Toledo Co., Ltd., Shanghai, China) according to the instructions. The probe was washed with distilled water before and after pH measurement, and the pH was measured three times to eliminate errors.

#### 2.2.6. Content Determination

The 0.1 mL of three different batches of ABZ nanosuspension were accurately transferred, dissolved in N, N-Dimethylformamide (DMF) with the aid of ultrasound for 5 min and diluted with mobile phase to prepare a test solution. The sample, after filtration through a 0.22-micron filter, was determined using high performance liquid chromatography (HPLC, Waters 2695 series), and the actual content was calculated using standard curve.

### 2.3. Solubility

The ABZ nanosuspension was lyophilized with a freeze dryer (Labconco, MO, USA) for 48 h to determine the saturation solubility of nanocrystals. Excess lyophilized ABZ nanocrystals and native ABZ were added into 20 mL of 25 ± 5 °C double distilled water, methanol, acetonitrile and DMF and then achieved dissolution equilibrium while oscillating in the shaker at 25 °C for 72 h (17). The supersaturated solution was centrifuged at 10,000 r/min (Hitachi Centrifugation CR21 GIII; Hitachi Koki Co., Ltd., Tokyo, Japan) for 10 min, and then the supernatant was collected for analysis using HPLC after filtration through a 0.22-micron filter.

### 2.4. In Vitro Dissolution

According to the guidelines of Veterinary Pharmacopoeia of the People’s Republic of China 2020, in vitro release of ABZ nanosuspensions was performed in PBS at pH = 2 and pH = 8, respectively, using a paddle dissolution method (dissolution tester, RC806, Tianjin Tianda Tianfu Technology Co., Ltd., Tianjin, China). The 50 μL of ABZ nanosuspension (containing 5 mg of ABZ) and 5 mg of ABZ bulk drug were injected into a dissolution vessel containing 1000 mL of PBS (pH 2.0, 8.0) solution preheated at 37 °C, respectively. The release medium after adding of drugs was placed at a temperature of (37 ± 0.5) °C at a speed of 60 r/min, and a certain volume (1 mL) of test solution was taken at 5, 15, 30, 60, 120, 240 and 360 min. The same volume of fresh PBS solution was re-added to maintain a constant volume at each collect time point. The sample solution was filtered by 0.22 μM filter and then injected into HPLC to determine the concentration of ABZ. Then cumulative release percentage of ABZ nanosuspension and native ABZ vs. time was plotted.

### 2.5. Pharmacokinetics Study

The pharmacokinetics experiment was performed according to “Guidelines of the Care and Use of Laboratory Animals” of Huazhong Agricultural University and approved by the Ethics Committee of Huazhong Agricultural University. Twelve healthy SD rats (half male and half female) with a uniform weight (300 ± 50 g) were used to evaluate the pharmacokinetics of ABZ nanosuspension. The rats were randomly divided into 2 groups with 6 rats in each group. The rats were deprived of food and water for 12 h before the start of experiment. Each group was randomly orally administered the prepared ABZ nanosuspension and bulk ABZ at the dose of 100 mg/kg, respectively. At the fixed time points of 0.5, 1, 2, 4, 6, 8, 12, 24, 36, 48, 72 and 96 h after administration, blood samples (0.3 mL) were taken from heart into 1.5 mL centrifuge tube. The plasma was obtained by centrifugation at 4000 g/min for 2 min after standing for 30 s and then frozen in −20 °C refrigerator before usage.

### 2.6. HPLC Assay

All the samples were analyzed using HPLC (Waters 2695 series) combined with an ultraviolet detector system with the wavelength set at 295 nm. For in vitro study, the chromatographic separation was achieved with an analytical ZORBAX SB C18 column (250 × 4.6 mm, i.d. 5 µm; Agilent Technology, Palo Alto, CA, USA) at 30 °C. The mobile phase was 0.5% acetic acid solution and acetonitrile. The flow rate and injection volume were 1 mL/min and 50 μL, respectively. The chromatograms of ABZ, ABZSO and ABZSO2 are shown in Figure 3. Linearity of the standard curves for ABZ and its metabolites of ABZSO and ABZSO2 ranged from 0.025 to 8.0 µg/mL, and the correlation coefficient (r) values were >0.9994. The limit of detection (LOD) and the limit of quantitation (LOQ) were 0.05 µg/mL and 0.1 µg/mL, respectively. Inter- and intraday relative standard deviations were <3.0%.

For plasma sample treatment, 100 µL of 0.1 mol/mL ammonia solution were added into 100 μL serum sample under vortex. After shaking for 1 min, 2 mL acetonitrile and 2 mL ethyl acetate were added and then further vortex for 1 min followed by ultrasonication for 5 min. The mixture was centrifuged for 10 min at 5000 r/min, and the precipitate was re-extracted by adding 2 mL ethyl acetate. The mixed extraction solution of two times was dried under nitrogen at room temperature, and the residue was dissolved with 200 μL methanol. The resuspension solution was centrifuged for 1 min at 4000 g (Hitachi Centrifugation CR21 GIII; Hitachi Koki Co., Ltd., Tokyo, Japan), and the supernatant, after filtration through a 0.22-micron filter, was taken for HPLC analysis. Separation of chromatographs was achieved using an Agilent Eclipse XDB-C18 column (250 mm × 4.6 mm i.d., 5 µm) at a flow rate of 1 mL/min at 30 °C. The gradient elution mode was set for separation of ABZ and its main metabolites, with the mobile phase A containing 0.5% acetic acid solution and mobile phase B containing pure acetonitrile with gradient elution. The gradient elution procedure was set as follows: 0 min, 90% A; 8 min, 25% A; 12 min, 90% A; and 15 min, 90% A. The correlation coefficient (r) values of ABZ, metabolite ABZSO and ABZSO2 at different concentrations (0.05 μg/mL, 0.1 μg/mL and 0.5 μg/mL) were >0.9994. The LOD and LOQ of ABZ, ABZSO and ABZSO2 were 0.05 μg/mL and 0.1 μg/mL, respectively. The mean plasma recovery rates of ABZ, ABZSO and ABZSO2 were 97.89–112.09%, 98.06–119.11% and 95.82–99.18%, respectively.

### 2.7. Statistical Analysis

The data were analyzed via the one-way analysis of variance (ANOVA) using the SPSS 17.0 Windows program (SPSS Co., Chicago, IL, USA). Significant differences and extremely significant differences were determined as *p* values of 0.05 and 0.01, respectively.

## 3. Results

### 3.1. Effect of Cosolvent on the Stability of ABZ Nanosuspension

According to the principle of neutralization and solubility of ABZ in acid and alkali, saturated malic acid, citric acid, diethylene glycol ether (DECS) and triethylamine were selected. It was found that it took a very high temperature (90 °C) to completely dissolve ABZ when DECS and triethylamine were used as cosolvents. At the same time, unknown impurities were detected using HPLC, compared to pure ABZ. The results show that ABZ was not stable at the high temperature of 90 °C (Figure 4). In comparison to citric acid, it took less time for ABZ to completely dissolve in malic acid at the temperature of 80 °C. Therefore, malic acid was selected as the cosolvent.

### 3.2. Effect of Stabilizer on the Stability of ABZ Nanosuspension

According to the early results of our data, polyvinyl pyrrolidone (PVP_k30_), polyvinyl alcohol (PVA), Tween 80 and hydrogenated castor oil with polyoxymethylene (HEL-40) were selected as stabilizers to prepare the nanosuspension. Four different surfactants with 5% concentration and saturated malic acid as cosolvent were used to prepare the nanosuspension. It was found that the suspensions with PVP_k30_ as surfactant had obvious stratification phenomenon, while the nanosuspension showed different degrees of agglomerations and poor mobility when using HEL-40. The nanosuspension prepared using PVA and Tween 80 was a uniform milky white suspension and hardly produced sedimentation (Figure 5). When sanding for 1 week, the nanosuspension stabilized with PVA still did not produce stratification and sedimentation. Therefore, PVA was selected as the stabilizer for this test.

### 3.3. Properties of ABZ Nanosuspensions

The optimization formulation condition and properties of ABZ nanocrystal is shown in Table 1. ABZ nanosuspension was a uniform suspension of a milky white color. Under SEM, the shape of the nanocrystals were elongated and evenly dispersed (Figure 6). In addition, the average hydration size of nanocrystals was 508 ± 11.869 nm with a PDI of 0.28 ± 0.01 and zeta potential of 2.30 ± 0.44 mV. ABZ contents and sedimentation rate as well as pH values of the nanosuspension were 10%, 1 and 3.5, respectively. The recovery rates of ABZ from ABZ nanocrystals were 98.5–102.3%. The redispersion time was only 27 ± 1.0 s in the case of the layered ABZ suspension under the magnetic shaker at a rotating rate of 20 r/min.

### 3.4. Identification

The retention time of ABZ nanocrystals and ABZ standard was determined to be the same using HPLC. There was no impurity peak in the drug extracted from the nanocrystals. It showed that the properties of ABZ nanocrystals prepared by this process were basically unchanged. The DSC thermogram of ABZ showed that there was a unique endothermic peak of ABZ at 213 °C (Figure 7). The nanocrystals showed a characteristic melting peak of ABZ at 211 °C, and the height of the endothermic peak did not change compared to the native drug, which indicated that the crystal form and crystallinity of ABZ in the nanosuspensions did not change compared to the native ABZ. Malic acid showed a characteristic peak at 110 °C, while the characteristic peak vanished in the nanocrystals. This might be due to the fact that malic acid is present in molecular form in the nanocrystals. The XRD pattern of the native drug and prepared ABZ nanocrystals are shown in Figure 8. The XRD intensity decreased at 2 theta of “11” and “18”, and the sharp peaks disappeared between 2 theta “20–22”. This difference might be due to the inference of the malic acid and PVA since only 10% of the drug was present in the lyophilized powder of the nanocrystals. A reduction in peak intensity after nanocrystallization was also reported [24]. The disappeared sharp peaks might be due to the relative crystallinity transformation of ABZ nanocrystals covered by PVA. As mentioned in the previous literature, ABZ has two polymorphs (Forms I and II). Form II is enantiotropically related to Form I. As shown in Figure 8, ABZ might crystallize into the thermodynamically stable Form II in nanocrystals NC systems [23].

### 3.5. Equilibrium Solubility of ABZ Nanosuspension

The saturated solubility of ABZ nanocrystals in water and different organic solvents was 2.2~118.3 fold higher than that of native ABZ, as shown in Table 2. The solubility of ABZ nanocrystals in water was 396.172 ± 0.053 μg/mL, which was enhanced 118.3 times compared to the solubility of native ABZ in water (3.349 ± 0.098 μg/mL).

### 3.6. In Vitro Dissolution

In vitro dissolution of native ABZ and ABZ nanocrystals is shown in Figure 9. Compared to native ABZ, the release rate and release degree of ABZ nanocrystals at pH = 2 and pH = 8 were significantly increased. When pH = 2, the ABZ nanocrystals released 98.1% of the drug within 5 min, while native ABZ only released 21.0%. When pH = 8, the ABZ nanocrystals released 85.8% of the drug within 5 min and nearly completely released the drug at 30 min, while native ABZ only released 20.5% at 5 min and was close to the maximum value of 87.9% at 2 h.

### 3.7. Pharmacokinetics of ABZ Nanocrystals

The plasma concentrations of the ABZ prototype and its active metabolites (ABZSO and ABZSO2) vs. time curves for ABZ nanocrystals and native ABZ after gastric administration are shown in Figure 10. The peak concentration of ABZ prototype in nanocrystals and the native ABZ group rapidly reached 0.66 ± 0.19 μg/mL at 0.5 h and 0.61 ± 0.06 μg/mL at 0.60 ± 0.22 h, respectively. They then slowly decreased and were lower than the detection limit after staying in plasma for 96 and 72 h, respectively. The ABZSO, the first metabolite of ABZ, reached the peak concentration of 2.08 ± 0.38 μg/mL at 4.67 ± 1.03 h and 1.18 ± 0.16 μg/mL at 5.00 ± 1.15 h in the nanocrystals and native ABZ groups, respectively. The second metabolite ABZSO2 in the nanocrystals and native ABZ groups reached the peak concentrations of 1.71 ± 0.30 μg/mL at 7.33 ± 1.03 h and 1.03 ± 0.11 μg/mL at 8.00 h, respectively. Both metabolites in the native ABZ and nanocrystals groups gradually decreased and they stayed in the plasma for more than 96 h. The total compounds in native ABZ and ABZ nanocrystals groups reached the peak concentrations of 2.08 ± 0.35 μg/mL and 3.37 ± 0.82 μg/mL at 5.60 ± 1.67 and 5.67 ± 1.51 h, respectively. The peak concentration of the total compounds of the ABZ nanocrystals was 1.62 fold that of the native ABZ.

The pharmacokinetic parameters are listed in Table 3. The absorption of ABZ nanocrystals was increased compared to that of the native ABZ. The area under the concentration–time curve (AUC0-∞), mean residence time (MRTlast) and elimination half-life (T1/2) of total compounds in the ABZ nanosuspension group were 101.72 ± 20.83 h*μg/mL, 32.98 ± 3.33 h and 83.73 ± 22.96 h, respectively, which were 1.40 fold (72.46 ± 6.18 h*μg/mL), 1.02 fold (32.02 ± 3.00 h), 1.04 fold (79.91 ± 12.65 h) compared to the native ABZ group, respectively. The AUC0-∞, MRTlast and T1/2 of the ABZ prototype in nanocrystals group were 16.38 ± 1.8 h*μg/mL, 43.24 ± 4.15 h and 66.78 ± 10.77 h, respectively, which were 1.05, 1.23 and 2.02 times higher than those of the native ABZ group with 14.66 ± 4.04 h*μg/mL, 35.17 ± 3.37 h and 62.40 ± 7.88 h, respectively. The AUC0-∞, MRTlast and T1/2 of ABZSO in the ABZ nanocrystals group (46.05 ± 11.08 h*μg/mL, 34.28 ± 2.10 h, 75.79 ± 21.68 h) increased by 1.46 fold (31.58 ± 7.49 h*μg/mL), 1.11 fold (30.88 ± 4.37 h) and 1.13 fold (67.31 ± 8.24) compared to the native ABZ group, respectively. The AUC0-∞, MRTlast and T1/2 of ABZSO2 in the nanocrystals group were 1.17, 1.02 and 1.03 fold than those of the native ABZ group, respectively.

## 4. Discussion

Poor oral absorption and low concentration in the blood and liver seriously affect the therapy efficacy of ABZ due to its insolubility [28,29]. It is reported that nanocrystalline technology can reduce the drug particle size to the nanosized range and thus increase the dissolution and bioavailability of insoluble ABZ by increasing its specific surface area. Many nanocrystals with unique physicochemical properties have been produced as a promising drug delivery system for ABZ (Table 4). The reported production method mainly included HPH combined with spray-drying processes and an antisolvent precipitation technique combined with drying, which are not suitable for scale production. For the HPH method, contamination due to the long duration of homogenization and high energy production are the main disadvantages, which directly influence the crystallinity of nanocrystals, and it is often used in the laboratory. The antisolvent precipitation method is a classical bottom-up procedure to formulate nanocrystals, employing precipitation by solvent–antisolvent addition and precipitation using solvent removal [18]. As an organic solvent is employed during this process, it is not an eco-friendly fabrication method.

In order to promote clinical application, the ABZ nanocrystal suspension was prepared using a novel bottom-up approach based on acid-base neutralization followed by high-speed mixing and dispersing to easily achieve industrial production. Firstly, the ABZ was dissolved in malic acid solution at 80 °C and then rapidly stirred. In the presence of surfactant, the temperature was reduced, and the same molar amount of cold sodium hydroxide solution was added at the same time. The neutralization of acid and alkali resulted in the sudden supersaturation of ABZ in weakly acidic water and the formation of fine grains. Then the ABZ nanocrystals was treated using high-speed mixing and dispersing to get uniform nanocrystals. In this process, it is necessary to control the dissolution temperature of ABZ to prevent its decomposition from producing impurities. At the same time, as the total surface area of the prepared nanocrystals particles is several orders of magnitude larger than that of the coarse particles, a large number of surfactant molecules needed to be added to prevent the aggregation of drug crystals so as to ensure the stability of nanocrystals [30,31]. It is important to select a suitable stabilizer because it plays a key role in the behavior of NCs as well as their stability. Therefore, the selection of cosolvent and surfactant was very important in the preparation of ABZ nanocrystals. The choice of cosolvent was selected according to the solubility and stability of ABZ. ABZ was dissolved quickly at 80 °C, and no impurities were generated. Therefore, 80 °C was chosen as the dissolution temperature. Therefore, among the four selected cosolvents, the saturated malic acid with the lower melting point was preferred. The selection of surfactants is mainly influenced by the similar hydrophobicity between drugs and surfactants. It is reported that similar hydrophobicity can provide better spatial stability [32,33,34,35,36]. In addition, the changes in pH and surface charge will also destroy the stability of the nanocrystals, leading to crystal aggregation. It is reported that ion stability with high zeta potential cannot guarantee the stability of nanocrystals and that the surface neutral surfactant can enhance the stability of drug nanocrystals. Among the four surfactants of PVPk30, PVA, Tween 80 and HEL-40 selected in this experiment [37], PVA has the longest stabilizing effect, and thus it was selected as the best surfactant in this experiment.

A high-speed dispersing device can effectively reduce particle size and polydispersity [38,39,40,41]. In this study, the best speed and stirring time were optimized. It was found that the nanocrystal could obtain uniform nanoparticles without obvious agglomeration by stirring for 5 min under 2000 rpm. It is reported that the small size of nanoparticles is more conducive to drug absorption and stability [33]. The DSC thermogram showed that the melting peak and the peak height of ABZ nanocrystals was the same as the endothermic temperature of ABZ. The HPLC and XRD determination also showed that the molecular structure was stable, and crystal transformation of the nanosuspension was observed after recrystallization. These results showed that the preparation process had no adverse effect on the drug structure. The saturated solubility of ABZ in different solvents significantly increased when ABZ was prepared into nanocrystals. The increase in saturated solubility may be due to the large increase in specific surface area and the decrease in particle size, resulting in the large increase in dissolution rate [42]. The dissolution rate of the ABZ nanocrystals also significantly increased. It is reported that the dissolution rate will be very fast when the particle size is less than 1 μm [33].

ABZ can be rapidly metabolized into ABZSO in vivo, and then ABZSO can be metabolized into ABZSO2, among which ABZSO has the highest blood concentration. It has been reported that ABZSO2 has a similar effect to ABZSO on intestinal and nonintestinal trichinella spiralis, but ABZSO2 has no effect on Echinococcus granulosus [43]. At the same time, most of the literature only detected the metabolite of ABZSO and no ABZ prototype and ABZSO2. In this study, ABZ and its two metabolites were all determined, and the respective pharmacokinetic parameters of each active ingredient and the total of the three active ingredients were calculated. The results show that the ABZSO concentration was the highest in both the native ABZ and ABZ nanocrystals groups, which is consistent with the previous literature [44,45]. The time to reach peak concentration of ABZ, ABZSO and ABZSO2 in the nanocrystals group and the total compounds was earlier for all than that of the native ABZ, and the maximum blood concentration was also higher than that of the native ABZ group. Compared to the native ABZ group, the bioavailability of ABZ, ABZSO, ABZSO2 and the totals of the three active compounds in the nanocrystals group increased by 1.14, 172, 1.12 and 1.40 fold, respectively. The increased bioavailability and absorption rate may be due to the decrease in particle size and the increase in saturated solubility and dissolution rate. The huge specific surface area of nanocrystals with small particle sizes will increase the adhesiveness of drugs to the gastrointestinal mucosa, thus increasing their absorption and residence time in the gastrointestinal tract [46]. In future research, the size of nanocrystals should be further reduced to more significantly enhance oral absorption. All the above data show that the preparation of nanosuspension using acid-base neutralization recrystallization combined with a high-speed dispersion method might be a promising way to enhance the solubility and oral bioavailability of ABZ.

## 5. Conclusions

In this study, ABZ nanocrystals were prepared using a bottom-up method based on acid-base neutralization recrystallization with high-speed mixing and dispersing. The cosolvent, stabilizer and temperature were optimized using a single factor test. This process is easy to scale up as it can be done in close proximity and could reduce the chances of contamination. Additionally, as no organic solvent is employed during this process, it could be an eco-friendly fabrication method. Simple instruments, low energy, less heat generation and low cost are some characteristic features of this method. This technique could overcome the problems associated with top-down and bottom-up approaches. The HPLC, DSC and XRD demonstrated that the molecular structure and crystal form remain unchanged in the preparation process. The saturation solubility and dissolution were significantly enhanced by nanocrystallization, and the bioavailability of ABZ was significantly improved. The nanosuspension prepared by acid-base neutralization recrystallization combined with high-speed dispersion method might be a promising way to enhance the solubility and oral bioavailability of ABZ and other insoluble drugs.

## Figures and Tables

**Figure 1 nanomaterials-12-03032-f001:**
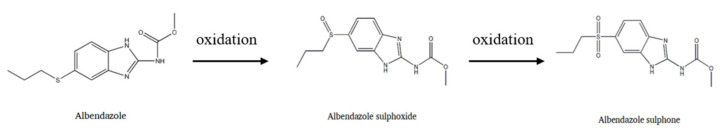
Albendazole metabolic pathway.

**Figure 2 nanomaterials-12-03032-f002:**
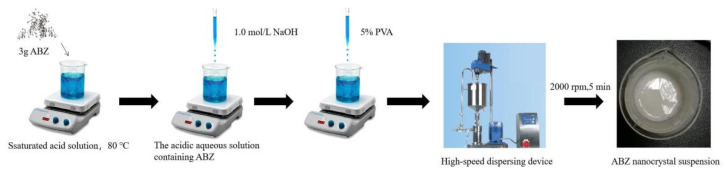
The schematic representation of ABZ nanocrystals preparation.

**Figure 3 nanomaterials-12-03032-f003:**
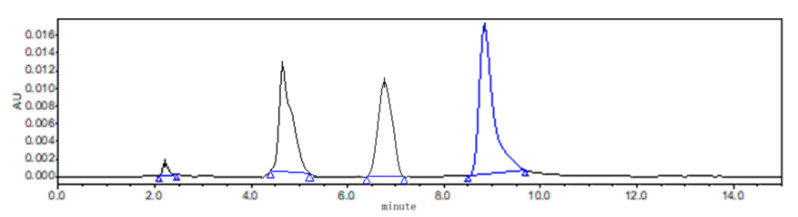
The chromatogram of ABZ, ABZSO and ABZSO2.

**Figure 4 nanomaterials-12-03032-f004:**
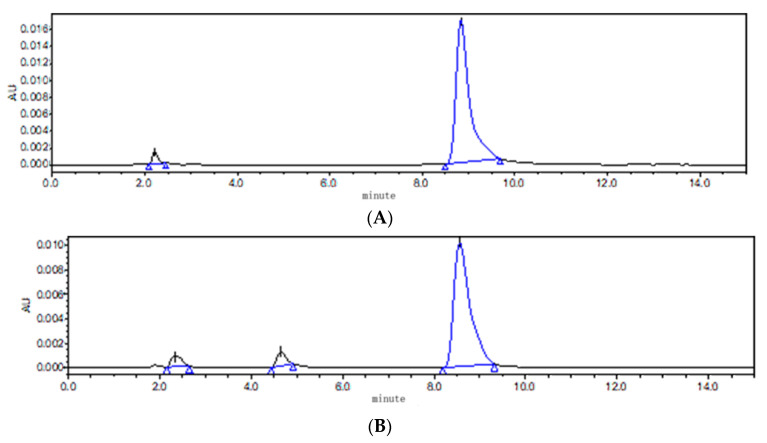
The chromatogram of ABZ (**A**) and ABZ (**B**) under high temperature of 90 °C.

**Figure 5 nanomaterials-12-03032-f005:**
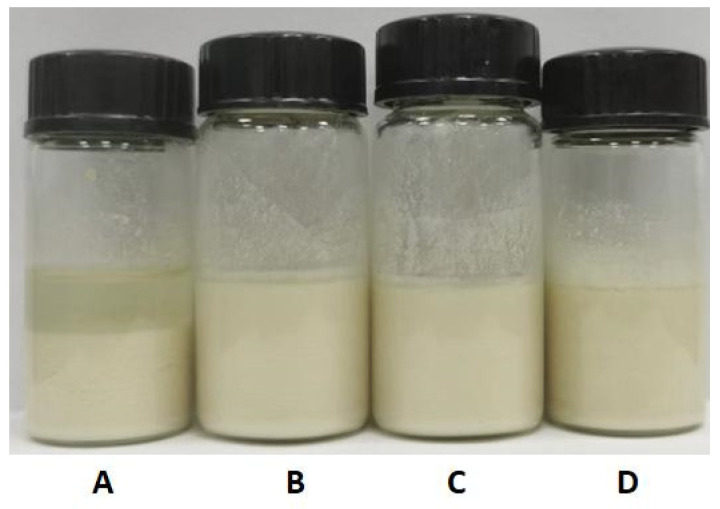
The appearance of the nanosuspension prepared by different stabilizers. (**A**): PVP, (**B**): PVA, (**C**): Tween 80; (**D**): HEL-40 as stabilizers.

**Figure 6 nanomaterials-12-03032-f006:**
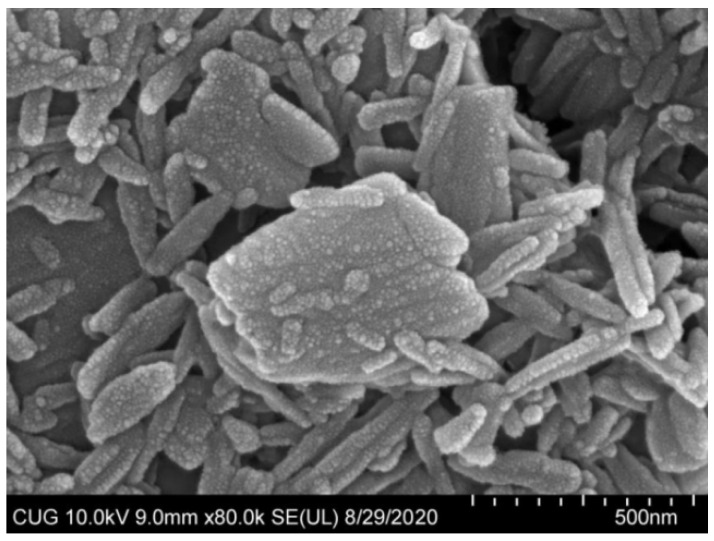
SEM photomicrograph of albendazole nanoparticles (×100,000).

**Figure 7 nanomaterials-12-03032-f007:**
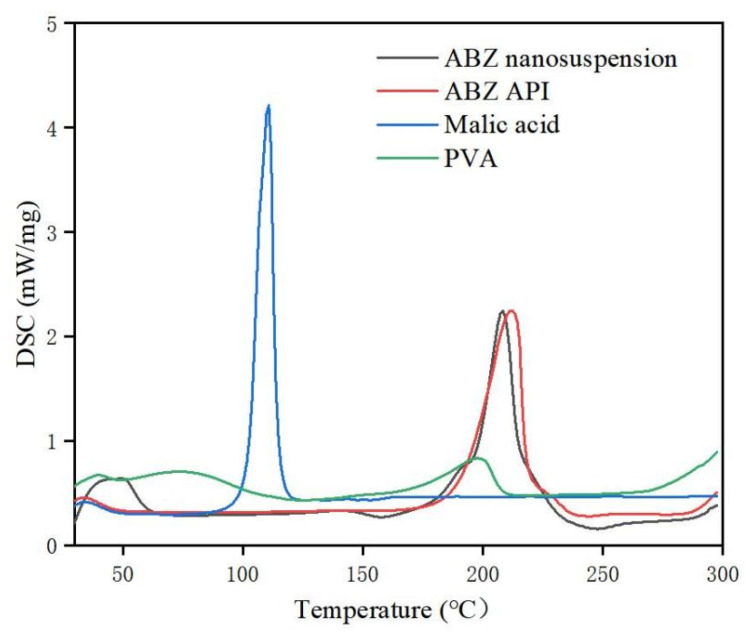
DSC of bulk drug and albendazole nanocrystals.

**Figure 8 nanomaterials-12-03032-f008:**
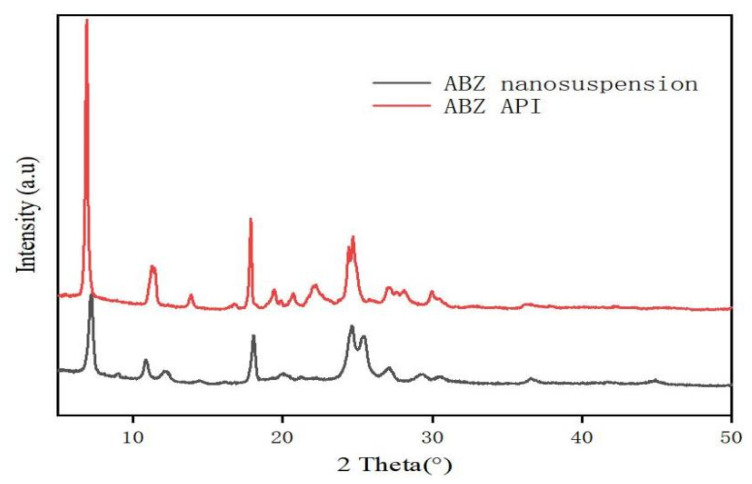
XRD pattern (B) of bulk drug and albendazole nanocrystals.

**Figure 9 nanomaterials-12-03032-f009:**
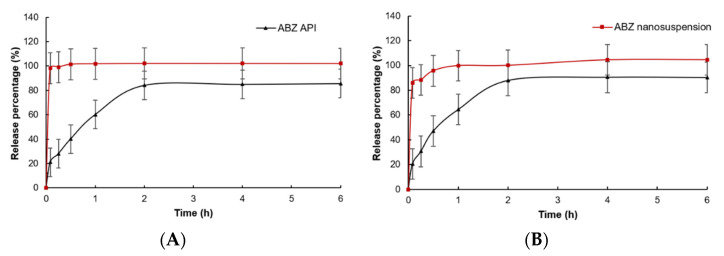
In vitro dissolution curves of native albendazole and albendazole. Nanocrystals in (**A**) pH = 2.0 PBS and (**B**) pH = 8.0 PBS (Mean ± SD, n = 6).

**Figure 10 nanomaterials-12-03032-f010:**
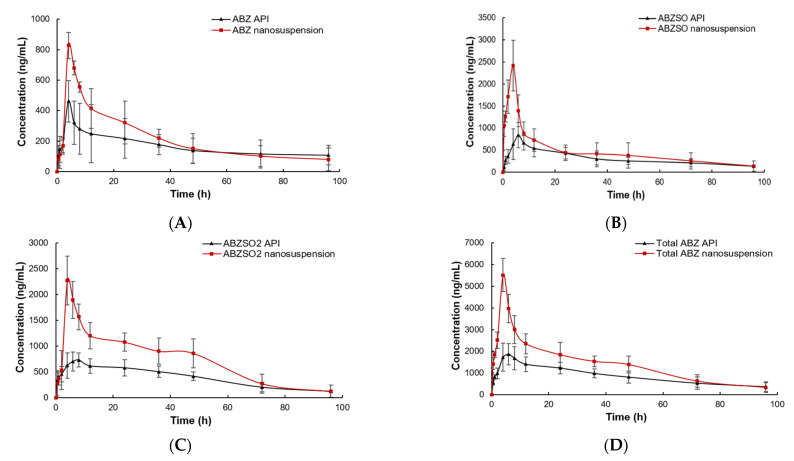
The concentration vs. time curves of ABZ (**A**), ABZSO (**B**), ABZSO_2_ (**C**) and total active ingredient (**D**) in SD rats after oral administration of nanocrystals and native ABZ at a dose of 100 mg/kg body weight.

**Table 1 nanomaterials-12-03032-t001:** Optimization formulation condition and properties of ABZ nanocrystal.

ABZ	PVA	Malic Acid	Temperature	Speed	Size (nm)	ZP (mv)	PDI
3 g	0.25 g	15.0 g	80 °C	2000 rpm/5 min	508 ± 11.9	2.30 ± 0.44	0.28 ± 0.01

**Table 2 nanomaterials-12-03032-t002:** Solubility of native ABZ and ABZ nanocrystal in various solvents (25 °C, standard atmospheric pressure).

Solvents	Solubility (µg/mL, Mean ± S.D.)		Enhanced Folds
	Native ABZ	ABZ Nanocrystal	
Water	3.35 ± 0.10	3967 ± 0.05 ^b^	1189
Ethanol	11.7 ± 0.00	226 ± 0.08 ^b^	19.4
Acetonitrile	31.9 ± 0.30	282 ± 0.24 ^b^	8.80
DMF	7308 ± 0.34	16,078 ± 0.42 ^b^	2.20

^b^ Statistical significances compared to ABZ power are *p*  <  0.01.

**Table 3 nanomaterials-12-03032-t003:** Pharmacokinetic parameters of native ABZ and nanocrystals in SD rats after an oral dose of 100 mg/kg (mean ± standard deviation, n = 6).

Parameters	Units	ABZ		ABZSO		ABZSO_2_		Total	
		Native	Nanosuspension	Native	Nanosuspension	Native	Nanosuspension	Native	Nanosuspension
T_max_	h	0.60 ± 0.22	0.50	5.00 ± 1.15	4.67 ± 1.03 ^b^	8.00	7.33 ± 1.03 ^a^	5.60 ± 1.67	5.67 ± 1.51 ^b^
C_max_	μg/mL	0.61 ± 0.06	0.66 ± 0.19	1.18 ± 0.16	2.08 ± 0.38 ^a^	1.03 ± 0.11	1.71 ± 0.30 ^a^	2.08 ± 0.35	3.37 ± 0.82 ^b^
AUC_0→__∞_	h*μg/mL	14.7 ± 4.04	16.4 ± 1.81 ^a^	31.6 ± 7.49	46.1 ± 11.1 ^a^	30.7 ± 2.25	35.9 ± 3.59 ^a^	725 ± 6.18	102 ± 20.8 ^b^
VZ	mL/kg	341 ± 85.5	340 ± 77.9 ^b^	143 ± 14.2	104 ± 16.9 ^b^	125 ± 10.2	113 ± 21.6 ^b^	89,468 ± 34,099	41,904 ± 13,965 ^b^
CL	mL/h/kg	5.70 ± 1.81	3.42 ± 1.39 ^a^	3.62 ± 0.22	2.22 ± 0.76 ^a^	2.82 ± 0.52	2.35 ± 0.52 ^a^	1101 ± 521	926 ± 223 ^b^
MRT_last_	h	35.2 ± 3.37	43.2 ± 4.15 ^a^	30.9 ± 4.4	34.3 ± 2.10 ^b^	30.6 ± 2.59	31.3 ± 5.11 ^a^	32.0 ± 3.00	33.0 ± 3.33 ^a^
Ke	1/h	0.01 ± 0.001	0.01 ± 0.002 ^a^	0.010 ± 0.001	0.010 ± 0.003 ^b^	0.01 ± 0.002	0.008 ± 0.002 ^b^	0.009 ± 0.001	0.009 ± 0.002 ^a^
T_1/2_	h	62.4 ± 7.88	66.8 ± 10.8 ^b^	67.3 ± 8.24	75.79 ± 21.68 ^a^	87.82 ± 21.55	90.5 ± 26.4 ^b^	79.9 ± 12.7	83.7 ± 23.0 ^b^
F	%	114.7		172		113		140	

Tmax, time to reach maximum concentration; Cmax, maximum concentration of drug; AUC0_→∞_, area under plasma concentration–time curve; VZ, volume of distribution; CL, clearance rate; MRTlast, mean resident time; Ke, elimination rate; T1/2, elimination half-life; F, bioavailability. ^a^ Statistical significances compared to native ABZ are *p*  <  0.05. ^b^ Statistical significances compared to native ABZ are *p*  <  0.01.

**Table 4 nanomaterials-12-03032-t004:** The reported preparation methods of ABZ nanocrystal.

Method	Solvent and Stabilizer	Performance	Reference
High-pressure homogenization combined with spray drying	Water, PVPk30	Enhanced the chemoprophylactic and clinical efficacy	19
High-pressure homogenization combined with drying processes	Water, ABZ and P188 (1:1)	AUC values of nano-sized ABZO was enhanced by threefold	21
High-pressure homogenization combined with spray-drying processes	Water, P188	AUC of ABZSO was almost twice than that of the control	22
Spray drying	Dichloromethane, P188	Approximately 4.2-fold higher AUC of ABZSO in Beagle dogs	23
Antisolvent precipitation technique combined with drying	Dimethyl sulfoxide	Improved the dissolution	24
High pressure homogenization combined with and spray-drying	Water	The dissolution rate noticeably increased	25

## Data Availability

Not applicable.

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
