# Peer review of "Nanocrystal Suspensions for Enhancing the Oral Absorption of Albendazole"

_nanomaterials, 2022, doi:10.3390/nano12173032_

Round 1
Reviewer 1 Report
The manuscript is well written and experiments have been carefully described and carried out. Results are clearly presented.
Nonetheless, the originality of the study seems rather limited, and thus we encourage authors to better support and highlight the interest of their study and the advantages of the method used within the many already reported.
Author Response
Point 1: The manuscript is well written and experiments have been carefully described and carried out. Results are clearly presented. Nonetheless, the originality of the study seems rather limited, and thus we encourage authors to better support and highlight the interest of their study and the advantages of the method used within the many already reported.
Response: Thanks for you kind suggestions. The originality, difference and advantages of the production method was emphasized in the revised manuscript.
Many nanocrystals with unique physicochemical properties have been produced as a promising drug delivery system for ABZ (Table 3). The reported producation method mainly include HPH combined with spray-drying processes and anti solvent precipitation technique combined with drying, which are not suitable for scale production. For HPH method, contamination due to the long duration of homogenization and high energy production are the main disadvantages, which directly influence the crystallinity of nanocrystals and is often used in the laboratory. The antisolvent precipitation method is a classical bottom-up procedure to formulate nanocrystals, employing precipitation by solvent-anti-solvent addition and precipitation by solvent removal (18). Because the organic solvent is employed during this process, it was not an eco-friendly fabrication method. In this study, the ABZ nanocrystal suspension were prepared by a novel bottom-up approach based on the acid-base neutralization followed by high-speed mixing and dispersing to easily achieve the industrial production. This process is easy to scale up as it can be done in a close proximity and could reduce the chances of contamination. Additionally, because no organic solvent is employed during this process, it could be an eco-friendly fabrication method. Simple instrument, low energy, less heat generation and low cost are some characteristic features of this method. This technique could to overcome the problems associated with top-down and bottom-up approaches.
Bsesides, ABZ can be rapidly metabolized into ABZSO in vivo, and then ABZSO can be metabolized into ABZSO2, among which ABZSO has the highest blood concentration. It has been reported that ABZSO2 has similar effect to ABZSO on intestinal and non-intestinal trichinella spiralis. Most of the literature only detected the metabolite of ABZSO but no ABZ prototype and ABZSO2. In this study, ABZ and its two metabolites were all determined, the respective pharmacokinetic parameters of each active ingredient and the total of the three active ingredients were calculated.
Reviewer 2 Report
I would like to thank the authors for well designed and presented method. O suggest several minor improvements:
1. English editing
2. Updated references
3. Review the instruments models
Author Response
Point 1: English editing.
Response 1: The manuscript was thoroughly revised.
Point 2: Updated references
Response 2: Some references were added.
Point 3: Review the instruments models
Response 2: The instruments models were added.
Reviewer 3 Report
Section 2 should be presented more “technically”. Please apply. Also, please merge the characterization techniques presented in sections 2.2-2.7, while describing equipment used in the experiment – work development environment / work apparatus should be given – model of equipment (manufacturer, city, country). Section 2.7 Please detail “content”. Section 2.9. line 139 – please introduce the appropriate reference.
Results. Figures 5 and 6. The title are not correct.
Results and Discussion. In current form, the level of section is weak to moderate and the manuscript seems to be only an enumeration of information/ obtained results. This issue should be corrected by highlighting the main findings, shortening the information presented, while keeping only the significant results and compare them with similar studies (please apply) in a more "technical" way. Indeed, there is a lot of work and results, but in this form, the manuscript is difficult to read and to remark the most important findings.
Please improve the quality of Figures 7 and 8. Figure 8 is of poor quality, making difficult any interpretation, remark. Please use the same color lines for the same sample through the entire manuscript.
When referring to values, please use the following format: 111, 11.1 and 1.11, respectively.
Finally, I consider that the paper is not proper for publication in the present format in the Nanomaterials journal. If the paper will not be improved, I will not endorse the publication. I would be happy to review an improved/ clear version of the manuscript.
Author Response
Point 1: Section 2 should be presented more “technically”. Please apply. Also, please merge the characterization techniques presented in sections 2.2-2.7, while describing equipment used in the experiment-work development environment/work apparatus should be given-model of equipment (manufacturer, city, country).
Response 1: Thanks for you kind suggestions. Sections 2.2-2.7 was merge as 2.2 2.2 Characterization of the ABZ nanocrystals. The instruments models were added.
Point 2: Section 2.7 Please detail “content”.
Response 2: It was revised. The detail HPLC assay method was described in section 2.6. HPLC assay.
Point 3: 3.Section 2.9. line 139 -please introduce the appropriate reference.
Response 3: The reference was added.
Point 4: Figures 5 and 6. The title are not correct.
Response 4: It was revised.
Reviewer 4 Report
Manuscript nanomaterials-1823005
The authors studied the nanosizing efficiency and improvement of the dissolution rate and oral absorption of albendazole.
Nanocrystals of albendazole were already described several times in the literature, as detailed in the introduction by the authors. The originality of the authors’ protocol (acid-base neutralization followed by high speed-mixing and dispersing) and its differences on experimental results are not sufficiently explained (despite a short paragraph in the introduction L52-69).
In my opinion, this work can be published in “nanomaterials” after major corrections.
Major correction 1: First, the authors need to add, in the discussion, a paragraph where they compare their protocol to the others described in the literature. Moreover, other protocols yield to albendazole nanocrystals which perhaps have different physical properties and / or increase of oral absorption / dissolution rates. The authors need to make a comparison (in, for example, a table) to explain the interest of their protocol.
Major correction 2: The authors have evaluated dissolution rate with PBS medium pH2 and pH8, justifying this protocol with reference 20 (Int J Pharm 2020, 585, 119501). This reference deals with ricobendazole, and the authors have used classical HCl 0.1 M medium. For albendazole human medicines, dissolution must be evaluated with 900 mL of HCl 0.1 M at 50 rpm (see USP or literature). The authors need or to explain and detail why they used 1000 mL of PBS at 60 rpm, or to make experiments with the correct protocol (to compare their results with the other works described in the literature).
Minor corrections:
P1L32 and Figure 1: Please, replace “sulfonated” by “oxidized again” and “sulphonation” by “oxidation”. In chemistry, a sulphonation yields to a SO3H group. When a sulfoxide becomes a sulfone, it is an oxidation.
P2L84: please define “PVA”
P4L165-173: please, add chromatograms of ABZ, ABZSO and ABZSO2 and the resolution values.
P5L203: please, add a chromatogram of ABZ with unknown impurities.
P6L236-237: Please, determine recovery rates (ABZ nanocrystals versus ABZ standard).
P7L246-247: the authors say that the XRD profile did not show any significant difference between the native ABZ and the nanocrystals. IMO, when I see XRD figure, intensity decreased for several peaks (for example at approx. 2 theta “11” and “18”) and sharp peaks disappeared (between approx. 2 theta “20-22”). The authors need to comment these differences, or to provide a better figure (if they want to prove there is no difference).
P9L309: the first sentence of the discussion is not comprehensible. Please, correct this sentence.
P11L385: please, provide a reference for ABZSO2 effects.
Typos:
P1L15: “evulated” must be “evaluated”
P2L52: “solubilty” must be “solubility”
P2L62, L64 and L67: the order of the references (25, 26, 24) is wrong. Please correct.
P5L175: “0.1 mol/mL ammonia solution of 100 µL” must be “100 µL of 0.1 mol/mL ammonia solution”
P5L180: “were” must be deleted
P5L181: 4000 g/min must be or 4000 g or 4000 rpm
P6L241: “211 boC” must be “211 oC”
P7L243: “cpmpared” must be “compared”
P7, Figures 5 and 6 were inverted (Fig 5 is XRD and Fig 6 is DSC)
Table 2 “natvie” must be “native”
P9L323: “was” must be “were”
P11L370: rmp must be rpm
Author Response
Point 1: The authors studied the nanosizing efficiency and improvement of the dissolution rate and oral absorption of albendazole.
Nanocrystals of albendazole were already described several times in the literature, as detailed in the introduction by the authors. The originality of the authors’ protocol (acid-base neutralization followed by high speed-mixing and dispersing) and its differences on experimental results are not sufficiently explained (despite a short paragraph in the introduction L52-69).
In my opinion, this work can be published in “nanomaterials” after major corrections.
Response 1: Thanks for you kind suggestions. The difference and advantages of the production method was emphasized in the revised manuscript.
Point 2: Major correction 1: First, the authors need to add, in the discussion, a paragraph where they compare their protocol to the others described in the literature. Moreover, other protocols yield to albendazole nanocrystals which perhaps have different physical properties and / or increase of oral absorption / dissolution rates. The authors need to make a comparison (in, for example, a table) to explain the interest of their protocol.
Response 2: Thanks for you kind suggestions. The difference and advantages of the production method was emphasized in the revised manuscript.
Many nanocrystals with unique physicochemical properties have been produced as a promising drug delivery system for ABZ (Table 3). The reported producation method mainly include HPH combined with spray-drying processes and anti solvent precipitation technique combined with drying, which are not suitable for scale production. For HPH method, contamination due to the long duration of homogenization and high energy production are the main disadvantages, which directly influence the crystallinity of nanocrystals and is often used in the laboratory. The antisolvent precipitation method is a classical bottom-up procedure to formulate nanocrystals, employing precipitation by solvent-anti-solvent addition and precipitation by solvent removal (18). Because the organic solvent is employed during this process, it was not an eco-friendly fabrication method. In this study, the ABZ nanocrystal suspension were prepared by a novel bottom-up approach based on the acid-base neutralization followed by high-speed mixing and dispersing to easily achieve the industrial production. This process is easy to scale up as it can be done in a close proximity and could reduce the chances of contamination. Additionally, because no organic solvent is employed during this process, it could be an eco-friendly fabrication method. Simple instrument, low energy, less heat generation and low cost are some characteristic features of this method. This technique could to overcome the problems associated with top-down and bottom-up approaches.
Bsesides, ABZ can be rapidly metabolized into ABZSO in vivo, and then ABZSO can be metabolized into ABZSO2, among which ABZSO has the highest blood concentration. It has been reported that ABZSO2 has similar effect to ABZSO on intestinal and non-intestinal trichinella spiralis. Most of the literature only detected the metabolite of ABZSO but no ABZ prototype and ABZSO2. In this study, ABZ and its two metabolites were all determined, the respective pharmacokinetic parameters of each active ingredient and the total of the three active ingredients were calculated.
Point 3: Major correction 2: The authors have evaluated dissolution rate with PBS medium pH2 and pH8, justifying this protocol with reference 20 (Int J Pharm 2020, 585, 119501). This reference deals with ricobendazole, and the authors have used classical HCl 0.1 M medium. For albendazole human medicines, dissolution must be evaluated with 900 mL of HCl 0.1 M at 50 rpm (see USP or literature). The authors need or to explain and detail why they used 1000 mL of PBS at 60 rpm, or to make experiments with the correct protocol (to compare their results with the other works described in the literature).
Response 3: The reference 20 was deleted. In this study, PBS was used to conveniently adjust different pH to investigate whether the dissolution rate of nanocrystals in the stomach and intestines was both significantly increased.
Point 4: P1L32 and Figure 1: Please, replace “sulfonated” by “oxidized again” and “sulphonation” by “oxidation”. In chemistry, a sulphonation yields to a SO3H group. When a sulfoxide becomes a sulfone, it is an oxidation.
Response 4: It is revised.
Point 5: P2L84: please define “PVA”
Response 5: It is revised.
Point 6: P4L165-173: please, add chromatograms of ABZ, ABZSO and ABZSO2 and the resolution values.
Response 6: It is added.
Point 7: P5L203: please, add a chromatogram of ABZ with unknown impurities.
Response 7: It is added.
Point 8: P6L236-237: Please, determine recovery rates (ABZ nanocrystals versus ABZ standard).
Response 8: It was added.
Point 9: P7L246-247: the authors say that the XRD profile did not show any significant difference between the native ABZ and the nanocrystals. IMO, when I see XRD figure, intensity decreased for several peaks (for example at approx. 2 theta “11” and “18”) and sharp peaks disappeared (between approx. 2 theta “20-22”). The authors need to comment these differences, or to provide a better figure (if they want to prove there is no difference).
Response 9: It was revised.
Point 10: P9L309: the first sentence of the discussion is not comprehensible. Please, correct this sentence.
Response 10: It was revised.
Point 11: P11L385: please, provide a reference for ABZSO2 effects.
Response 11: It was added.
Point 12: P1L15: “evulated” must be “evaluated”
Response 12: It was revised.
Point 13: P2L52: “solubilty” must be “solubility”
Response 13: It was revised.
Point 14: P2L62, L64 and L67: the order of the references (25, 26, 24) is wrong. Please correct.
Response 14: It was revised.
Point 15: P5L175: “0.1 mol/mL ammonia solution of 100 µL” must be “100 µL of 0.1 mol/mL ammonia solution”
Response 15: It was revised.
Point 16: P5L180: “were” must be deleted
Response 16: It was deleted.
Point 17: P5L181: 4000 g/min must be or 4000 g or 4000 rpm
Response 17: It was revised.
Point 18: P6L241: “211 boC” must be “211 oC”
Response 18: It was revised.
Point 19: P7L243: “cpmpared” must be “compared”
Response 19: It was revised.
Point 20: P7, Figures 5 and 6 were inverted (Fig 5 is XRD and Fig 6 is DSC)
Response 20: It was revised.
Point 21: Table 2 “natvie” must be “native”
Response 21: It was revised.
Point 22: P9L323: “was” must be “were”
Response 22: It was revised.
Point 23: P11L370: rmp must be rpm
Response 23: It was revised.
Round 2
Reviewer 3 Report
Thank you for your efforts to improve the quality of the article. The authors took into account all my concerns. I think in the present form it can be accepted for publication.
Reviewer 4 Report
The authors have corrected their manuscript as requested. Their work can be published in Nanomaterials.